# Was *R* < 1 before the English lockdowns? On modelling mechanistic detail, causality and inference about Covid-19

**Simon N. Wood**[1]*, **Ernst C. Wit**[2]

**1** School of Mathematics, University of Edinburgh, Edinburgh, United Kingdom, **2** Institute of Computing, Università della Svizzera italiana, Lugano, Switzerland

* simon.wood@ed.ac.uk

**Data Availability Statement:** All relevant data are within the manuscript and its Supporting information files.

**Funding:** The author(s) received no specific funding for this work.

## Abstract

Detail is a double edged sword in epidemiological modelling. The inclusion of mechanistic detail in models of highly complex systems has the potential to increase realism, but it also increases the number of modelling assumptions, which become harder to check as their possible interactions multiply. In a major study of the Covid-19 epidemic in England, Knock et al. (2020) fit an age structured SEIR model with added health service compartments to data on deaths, hospitalization and test results from Covid-19 in seven English regions for the period March to December 2020. The simplest version of the model has 684 states per region. One main conclusion is that only full lockdowns brought the pathogen reproduction number, *R*, below one, with $R \gg 1$ in all regions on the eve of March 2020 lockdown. We critically evaluate the Knock et al. epidemiological model, and the semi-causal conclusions made using it, based on an independent reimplementation of the model designed to allow relaxation of some of its strong assumptions. In particular, Knock et al. model the effect on transmission of both non-pharmaceutical interventions and other effects, such as weather, using a piecewise linear function, *b*(*t*), with 12 breakpoints at selected government announcement or intervention dates. We replace this representation by a smoothing spline with time varying smoothness, thereby allowing the form of *b*(*t*) to be substantially more data driven, and we check that the corresponding smoothness assumption is not driving our results. We also reset the mean incubation time and time from first symptoms to hospitalisation, used in the model, to values implied by the papers cited by Knock et al. as the source of these quantities. We conclude that there is no sound basis for using the Knock et al. model and their analysis to make counterfactual statements about the number of deaths that would have occurred with different lockdown timings. However, if fits of this epidemiological model structure are viewed as a reasonable basis for inference about the time course of incidence and *R*, then without very strong modelling assumptions, the pathogen reproduction number was probably below one, and incidence in substantial decline, some days before either of the first two English national lockdowns. This result coincides with that obtained by more direct attempts to reconstruct incidence. Of course it does not imply that lockdowns had no effect, but it does suggest that other non-pharmaceutical interventions (NPIs) may have

**Competing interests:** The authors have declared that no competing interests exist.

been much more effective than Knock et al. imply, and that full lockdowns were probably not the cause of *R* dropping below one.

## Introduction

In principle the inclusion of known mechanisms into models used for statistical inference should improve inference by reducing the bias caused by model misspecification. But there is a catch. What happens if the mechanisms are themselves described only in an approximate manner by ad hoc sub-models? It is then possible for the assumptions built into the sub-models to introduce substantial misspecification bias. The real world consequences of such bias could be substantial if the model is used to determine major public policies. This paper examines and re-implements the model of [1] to investigate the robustness of the inferences about Covid-19 lockdowns made using it. We show that key results are entirely dependent on strong but incidental assumptions introduced in the model formulation, and that relaxation of those assumptions effectively reverses the conclusions.

This may matter in assessing the effectiveness of lockdowns and other stringent blanket measures, which have consequences in addition to reducing viral spread. For example, they modify the evolutionary landscape for the pathogen in ways that seem unlikely to offer a selective advantage for milder strains (see S1 Code). Among mitigation measures full stay-at-home lockdowns are also particularly severe in terms of creating the economic shocks that may cause economic hardship and exacerbate inequality in the long term. In England economic hardship and inequality are associated with very substantial loss of life, as reviewed at length in [2]. We can not predict the actual future life loss that lock down effects will cause, but figures are available that at least indicate the scale of the risk. [2] includes a detailed assessment of the health effects that followed on from the economic shock of 2008, which at minimum constitute a health burden of some 9 million lost life years for the current UK population (based on the increase in the deprivation related life expectancy gap, although Marmot argues for a rather higher figure). For comparison, the extra life loss burden that a minimally mitigated Covid epidemic would have caused is estimated at around 3 million years [3]. The Bank of England characterises the economic shock from UK lockdown and other Covid suppression measures as the largest in 300 years, much larger than 2008. This suggests that lockdowns (and indeed other measures) carry a *risk* of substantial life loss, and that it is therefore important neither to overstate their clear benefits, nor neglect their downsides, if policy choices are to result in the imposition of measures that broadly minimise risk of life loss in the round (it is obviously facile to reduce the question to a binary choice between lockdown and do nothing). Recognising this, the UK government has made some attempt to assess possible negative health effects of the measures imposed [3], but, although acknowledging that the long term economic impacts on health are likely to be large, has not quantified them. Looking beyond the UK, to India, UNICEF has identified particularly large effects of containment measures, in particular associated with the period of the Indian lockdown from March 24th 2020 [4]: they estimate about 150,000 extra childhood deaths and 60,000 extra still births for India. Given the age profile of Covid deaths, this corresponds to a life year loss more than double that implied by the official Indian Covid death toll to date, and obviously far above the life years saved by the lockdown according to the Government of India/public health foundation of India estimates of 80,000 deaths avoided [5].

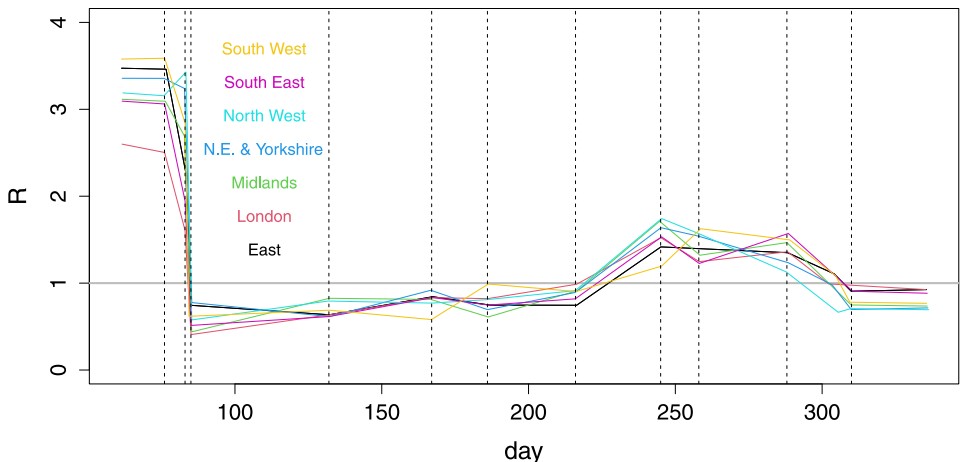

**Fig 1. Estimates of *R* by English region against day of year, as reported in** [1]. The plot is based on data digitized from Fig 1 of [1]. Uncertainties were not reported. The vertical lines mark model breakpoints at: 16th March movement restrictions (work from home advice), 23rd March lockdown announcement, 25th March 'Lockdown in full effect', May 11th initial easing, June 15th shops re-open, July 4th restaurants re-open, August 3rd eat-out-to-help-out scheme, September 1st schools open, September 14th rule of 6, October 14th Tier system, November 5th Lockdown. The kinks preceding November 5th are at a further model breakpoint. Prior to the first lockdown the following interventions occurred for which no breakpoints have been imposed: public information campaign, March 4th; symptomatic self isolation 13th; school and hospitality closures 20th. Full lockdown (stay at home orders and shutting down of much 'non-essential' activity) came into effect on 24th March, having been announced at 20:30 on 23rd March.

[1] is the 41st report of the COVID-19 response team from Imperial College London, whose reports have played a profound role in the shaping of UK government policy on Covid-19. Report 9 in the series provided a major component of the official justification for the first UK lockdown from March 24th 2020, and [1] was covered prominently in the UK *Sunday Times*, for example. A major message from [1] is that the pathogen reproductive number *R* was only reduced below one by full lockdowns in England in March and November (see Fig 1), with incidence apparently increasing until the eve of the March lockdown. We show that this result does not survive relaxation of some strong modelling assumptions. [1] also present 'counterfactual' simulations from the calibrated model from which they draw conclusions about the deaths that could have been avoided by an earlier first lockdown. We show that these simulations can not be viewed as 'counterfactuals' in the usual inferential sense (see e.g. [6]). The avoidable death figures are simple model extrapolations.

The model in [1] is an age-structured SEIR model with age-structured hospital compartments. The population is divided into 5-year age classes with a final 80+ class and two unstructured classes for care home residents and staff. There are 36 states in each of 19 classes (see Fig 2). The model was specified as a set of ODEs and converted to a discrete time stochastic model for fitting by the $\tau$-leap method [7]. The model was fitted to daily data on hospital deaths, care home deaths, hospital admissions, general ward occupancy, ICU occupancy, antibody test results and PCR test results from surveys, supplied as supplementary material for [1]. [1] also attempts to fit data on test results from the health system. However the model does not attempt to deal with the non-random, opportunistic nature of the sampling in this data stream, despite the continual changes in test capacity, criteria for testing, and operation of the contact tracing system over the course of the data. We therefore believe that there is substantial danger of these data simply undermining the analysis and they should not be included in data to be fitted (we made this decision at the outset, having concluded that we would strongly advice against

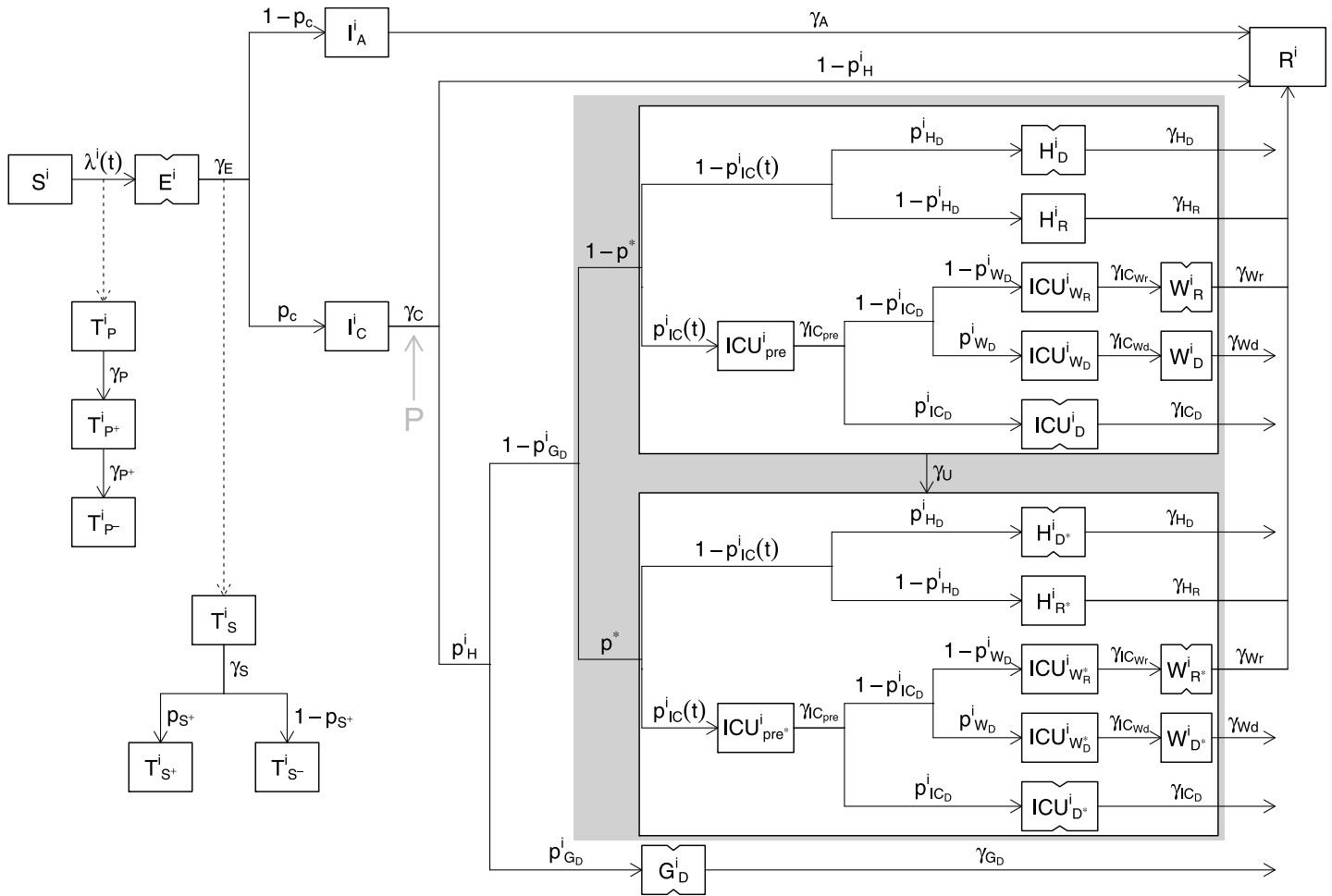

**Fig 2. Schematic diagram of the model compartments (boxes) and flows (arrows) for a single model age class, following supplementary Fig 2 of [1], but with the notational modifications used here, stages represented as two sequential compartments indicated with notched boxes, and the location of the extra stage $P$ that we insert to relax the generation time assumptions shown by the grey arrow and 'P'.** To obtain the rate of flow from one compartment to another, follow the path joining them in the direction of the arrow, multiplying the source state variable by the rate parameters labelling the segments of the path. Rates with a superscript $i$ vary with age class. The relative rates in different classes was obtained from a separate analysis reported in [1], with only a common multiplier of the class specific rates left as a free parameter. For example $p_H^i = p_H^{max} \psi_H^i$, where $\psi_H^i$ is fixed, but $p_H^{max}$ is free. Evaluation of original Knock et al. age-structured SEIR model and S1 Appendix A have full definitions.

use of these data if acting as statistical consultants, and have never attempted to fit these data). Data were available for seven English regions, which were fitted separately. The model has 26 free parameters.

[1] bases model inference (fitting) on particle filtering methods, with full fit to all regions reported to take over 100 CPU days, despite using only 96 particles per fit. This computational cost makes model checking difficult, particularly if a more usual number of particles is used and the stronger model assumptions are relaxed: the latter involves allowing substantially more free parameters plus hyper-parameters. Additionally [1] specifies massive overdispersion in all but the test data streams. Decreasing this over-dispersion to levels consistent with the data would likely increase particle depletion problems in filtering, leading to yet longer computing times. Given these issues, we will work directly with the ODE based model. The neglect of stochasticity in the state equations seems likely to be a minor issue here, relative to the other approximations made in the model. In particular, the only non-linearity in the model

dynamics is in the transmission between infectious and susceptible sub-populations, which contain large numbers except right at the epidemic start. Other model components are controlled by simple linear flows and are also aggregated over multiple age classes for fitting. Additionally the data sampling interval and total data duration are fairly short relative to the model's dynamic timescales. In any case, any results *dependent* on stochasticity would then require a much stronger justification for the stochastic formulation than that it was produced by discretisation of an underlying ODE model.

Furthermore, a generic strength and weakness of the particle filtering methods used in [1] is that they necessarily filter the state variables as well as model parameters. This is advantageous for state forecasting, but can be more problematic for inferential tasks. For an ill-specified dynamic model the filter is often forced to repeatedly select state transitions that are improbable under the model, in order to be sufficiently close to the data. This can result in the filtered states being in an extreme tail of the posterior predictive distribution of the model: that is, of the distribution implied by simulating unfiltered states from the model given the posterior distribution of parameters. Hence model adequacy needs to be checked by comparison of the data with simulations from the posterior predictive distribution. [1] does not report such checks, instead showing the filtered outputs. This is problematic when reality is then contrasted to 'counterfactual' simulations, necessarily from the posterior predictive distribution. The simple ODE approach used here does not filter. Instead the states are determined entirely by the model equations and the parameter values. This approach is unforgiving of model misspecification: adequacy is directly assessable from the model fit. It also reduces fit time by four orders of magnitude.

## Evaluation of original Knock et al. age-structured SEIR model

In this section we review the model of [1], before presenting some corrections and assumption relaxations in section Modification of the Knock et al. model. Fig 2 is a schematic showing the compartments in each 5-year age or care home class. The exposed, but pre-symptomatic, *E* stage is modelled by two sequential compartments. It is assumed that no infections are caused by this class. Symptomatic and asymptomatic stages $I_C$ and $I_A$ follow and cause infections, both are single compartment. The duration of the $I_C$ stage is set from data on time from onset of symptoms to hospital admission. The absence of pre-symptomatic infection will lead to longer generation times than are reported in the literature (e.g. [8, 9] p. 26), elevating the *R* estimates required to achieve observed epidemic growth rates. Care home residents are not hospitalised, and the $G_D^i$ class shown actually only receives patients for the care home resident class.

Model compartments for PCR and antibody test positivity are fed by the infection rate and the progression rate from the *E* state, respectively. The infection rate is driven by an age-structured mixing model with contact matrix, **C**, based on the POLYMOD survey data for the UK [10]. Most elements of **C** are multiplied by a function $b(t)$ modelling the impact of NPIs, and effects such as weather, on contact rates. In [1] $b(t)$ is piecewise linear with 12 breakpoints (and 12 free parameters) at policy change points. A major aim here is to relax the very strong assumptions built in to such a restrictive model. Care home contact rates are separately parameterized.

Hospitalized patients follow an ICU or general ward route. There are separate compartments for those eventually recovering or dying on the general ward. The ICU route has a pre-ICU compartment, from which patients enter compartments for those dying in ICU, entering ICU but dying later on the general ward, or entering ICU and recovering on the general ward. All compartments are duplicated for confirmed Covid (starred) and not yet confirmed (not

starred), with a parameter, $\gamma_U$, controlling the rate of testing based transfer from unconfirmed to confirmed. It is assumed that, from the start, 25% of patients arrive at hospital with confirmed Covid. This is improbable given initial testing capacity.

The model captures many features in impressive detail, but several aspects are not modelled:

1. Separation into locked down and key worker sub-populations at lockdown is not modelled, despite the very different values of *R* that must apply in these sub-populations, if lockdown is effective.

2. The assumed linearity of *b*(*t*) during lockdown precludes compensation for point 1 in fitting.

3. Seasonality or other non-NPI temporal effects on transmission are not modelled explicitly and are therefore confounded with the NPI effects, invalidating counterfactual manipulations of the latter.

4. Region-to-region transmission at the epidemic start is not represented, compromising early model fit and *R* estimates, as imported cases are modelled as local.

5. The assumption of no pre-symptomatic infectivity is inconsistent with empirical estimates of the serial interval and generation time, reviewed in [9], for example.

6. Within hospital transmission is not modelled, although hospital-acquired infections have been reported to account for a quarter of hospitalized cases at times in both waves [11], reports which are corroborated by public NHS data [12], and there is good evidence that the actual figure was higher [13]. This will compromise the hospital data fit.

7. No interaction between NPIs and age is allowed, which is unlikely given the risk-by-age profiles.

8. Differential transmission rates between symptomatics and asymptomatics are not modelled.

9. The reported differences in disease progression between men and women (see [14], for example) are not modelled.

10. Changes in testing rates with capacity changes are not modelled.

Any biological model for a complex system necessarily makes many simplifying assumptions, often without substantial detriment to statistical inference within the range of the data being modelled. However causal inference based on statistical methods puts much heavier requirements on the model, since it is then required to extrapolate. Counterfactual statements made using a model are of this causal character, and in the current case require the model to behave essentially as a mechanistic representation of reality (since we know of no causal inference strategy that could alleviate the effects of mis-specification in this sort of model, and [1] does not report any). Given this requirement for high mechanistic accuracy, any of the preceding omissions may be problematic. We note also that although we do not seek to extrapolate in this paper, most of these points will have some impact on our results. The hospital acquired infection issue makes it particularly difficult to exactly match hospital data with the model, for example.

## The basic SEI(R) model

For concreteness we describe the core of the SEIR model, giving the equations for other compartments in S1 Appendix A. Denoting the time derivative of a variable *x* by $\dot{x}$, then for

the ith class,

$$\dot{S}^i = -\lambda_i(t)S^i \tag{1}$$

$$\dot{E}^{i,1} = \lambda_i(t)S^i - \gamma_E E^{i,1} \tag{2}$$

$$\dot{E}^{i,2} = \gamma_E E^{i,1} - \gamma_E E^{i,2} \tag{3}$$

$$\dot{I}^i_A = (1 - p_c)\gamma_E E^{i,2} - \gamma_A I^i_A + \mathbb{I}(2 < i < 13)\phi_{t_0,\sigma_t}(t) \tag{4}$$

$$\dot{I}^i_C = p_c \gamma_E E^{i,2} - \gamma_c I^i_C. \tag{5}$$

$\lambda_i(t)$ is the force of infection defined below, and is the only interesting interaction between age classes. $p_c$ is the proportion of the infected showing symptoms, and the $\gamma$ parameters determine between compartment flow rates, given in [1]. $\mathbb{I}(\cdot)$ is an indicator function and $\phi_{t_0,\sigma_t}$ is an $N(t_0, \sigma_t^2)$ p.d.f. where $t_0$ is a free parameter. This initialization differs slightly from [1] who put 10 individuals in the age 15–20 asymptomatics at $t_0$. It is unclear why this is sensible, although it may slightly delay the first wave model care home epidemic. Susceptibles, $S^i$, are initialized from regional demography supplied in the [1] supplementary material. Care home sizes are supplied in the `sircovid` package by the `carehomes_parameters()` function [15].

The effective reproductive number of the pathogen, $R$, attempts to measure the number of new infections that each infected individual produces on average. Since this number obviously depends on the time course of the epidemic, there are various ways of defining it as an instantaneous quantity (see [9] for a review). For the current model structure the well established definition of [16] is appropriate, and ensures that $R = 1$ forms a sharp boundary between long term increase and decrease of the epidemic (that is, once $R$ falls below 1, long term decline is guaranteed until it exceeds 1 again). [1] uses this approach for each region, and we follow this. See S1 Appendix A.3 for details. Our fitting also requires the derivatives of the model states with respect to the parameters: the *sensitivities*. These follow directly from the model specification. For example if $S^i_{\theta_j}$ is the differential of $S^i$ w.r.t. $\theta_j$,

$$\dot{S}^i_{\theta_j} = -\frac{\partial \lambda_i}{\partial \theta_j}S^i - \lambda_i S^i_{\theta_j}.$$

Generically each term in the model equation involving a state gets replaced by that state's derivative w.r.t the parameter of interest, and to this are added any terms relating to direct dependence on the parameter of interest. For example, if $\gamma_C$ was a free parameter then $\dot{I}^i_{C_{\gamma_C}} = p_c \gamma_E E^{i,2}_{\gamma_C} - \gamma_c I^i_{C_{\gamma_C}} - I^i_C$. (Note that the same principle applies to the coefficients of the model component function $b(t)$ introduced below. $b(t)$ is represented using a basis expansion, and while the basis functions are time varying, the corresponding coefficients are not).

## Force of infection

Writing **I** for the vector of infectious individuals in each class, then the model for the force of infection in each class is $\lambda = \mathbf{MI}$ where

$$\mathbf{M} = \begin{pmatrix} b(t)\mathbf{C} & b(t)\mathbf{c}^{\text{chw}} & \epsilon b(t)\mathbf{C}_{\cdot,16} \\ b(t)\mathbf{c}^{\text{chw}} & m_{\text{chw}} & m_{\text{chw}} \\ \epsilon b(t)\mathbf{C}_{16,\cdot} & m_{\text{chw}} & m_{\text{chr}} \end{pmatrix}.$$

$\epsilon$, $m_{chw}$ and $m_{\text{chw}}$ are free parameters. $b(t)$ is a parameterized function of time controlling the variation of infection causing contact over time. **C** is a symmetric matrix of contact rates and **c**$^{\text{chw}}$ a vector (derived from it for carehome workers). $I_j$ is the sum of asymptomatic ($I_A^j$) and symptomatic infectious ($I_C^j$) in class $j$. S1 Appendix A.1 has the force of infection expressed so that sensitivities follow by inspection.

  **C** is based on the POLYMOD survey [10] accessed through the `socialmixr` R package [17]. This had 1011 UK participants, who each recorded their contacts on one day. There were 7 participants in the 75–80 age group and none over 80. S1 Appendix A.2 gives details.

## The likelihood

The likelihood is constructed from binomial components for the PCR and antibody test data (see S1 Appendix B.2), and negative binomial components for the hospital death, care home death, hospital admissions, general ward occupancy and ICU occupancy data. For the negative binomial components [1] sets $\kappa = \mu^2/(\sigma^2 - \mu)$ equal to 2 *in all cases* without justification offered. This is a huge level of overdispersion, heavily down-weighting the data relative to the priors. For example, hospital deaths show no evidence of over-dispersion relative to Poisson. But for an expected death rate of 200 the choice of $\kappa$ raises the standard deviation from 14, for a Poisson deviate, to 140. Although such a choice will reduce particle depletion problems in filtering, it is not easy to justify as a statistical model. Still more problematic is the assumption that observed daily bed occupancy is given by a negative binomial deviate with expectation given by the model, *with these deviates independent between days*. We are at a loss to understand what mechanism could give rise to such a model. A reasonable model might have daily arrivals and discharges as independent random variables with means given by the model, but occupancy obviously integrates these arrival and discharge rates over days, leading to strong dependence between days. The stochastic version of the model might model some of this dependence, but leaves even less justification for additional independent negative binomial variability.

## Modification of the Knock et al. model

In this section we present modifications of the Knock et al. model in order to deal with some of the deficiencies identified above. They consist of a number of corrections and minor modifications and, more fundamentally, relaxing some of the stronger modelling assumptions made in [1].

### Corrections and minor modifications

  **Rates.**  The $\gamma$ parameters controlling rates of progression between model compartments are either taken from the literature, or are estimated from CHESS (COVID-19 Hospitalisations in England Surveillance System) data that are not available for checking. There are at least two identifiable problems with the durations used in [1]. Firstly they set the mean duration of the *E* stage to 4.6 days citing [18]. That paper actually reports a mean of 5.5 days, with 4.6 days lying just above the lower 95% confidence limit for the median. Here we used the mean of 5.8 days from the meta-analysis of [19], which includes [18] as one of the studies. In fact the most statistically careful analysis we found [20] gives an estimated mean incubation period of 9.1 days (n = 1211), and generation time of 5–6 days. Secondly [1] assumes that the *mean* time from symptoms to hospitalization is 4 days based on [21], but that paper gives 4 days as the *median*. An exponential distribution is used for time from symptoms to hospitalization (a model which the figures reported in [21] do seem to support), so the median is log2 of the mean. Based on

the male and female medians of 5 and 4 days reported in [21], we therefore used a mean time to hospitalization of 6.5 days. In fact [21] is based on early data (up until April 19th 2020) from the ISARIC study. From the much larger ISARIC sample available by October 2020, the mean time from first symptoms to hospitalization is reported as 7.7 days [22], but we will nevertheless follow [1] in using Docherty et al., simply correcting the incorrect use of the median in place of the mean.

Another issue is the assumption that 25% of patients were arriving at hospital with a test confirming their status from the start of the epidemic. In fact, as documented in [23], there was no testing of patients outside of hospitals between 12 March 2020 and 28th April 2020, with very little capacity before this time and close to full testing capacity not reached until mid June 2020 (see Fig 1 of [23]). To crudely capture this we allowed $p^*$ to increase linearly from 0 to 0.25 between days 120 and 170, staying at 0.25 thereafter.

**Priors.** The priors used were not exactly those in [1], rather priors were set to be vague on a working parameter scale. Any limits on parameter were set by the prior intervals reported in [1]. Parameters were optimized on a working scale—either untransformed, log transformed or scaled logit transformed. Gaussian priors on the working scale were also applied, but except for $t_0$ these were vague, and their only purpose was to allow ready detection of any parameters that were not identifiable. See S1 Appendix B.1 for details.

## The negative binomial likelihood

While our basic conclusions are in fact unchanged if we use the likelihood given in [1] for the hospital occupancy data, we can see no valid justification for this part of the model formulation, and therefore replaced it with a likelihood based on the daily change in occupancy. In particular we model the ward (or ICU) arrivals and departures as independent overdispersed Poisson deviates, the difference in which gives the daily change in occupancy. A difficulty with applying this model directly is that hospital arrivals and discharges tend to have weekly pattern. This pattern shows up strongly in the ACFs and PACFs of occupancy first differences for some regions, especially east of England, but is absent from the model. We therefore base the likelihood on weekly changes. Since the changes in occupancy carry no information on the level of occupancy, we also add the sum of daily bed occupancies as a final datum to be fitted, treating this as close to Poisson (by setting $\kappa$ to a very high constant). See S1 Appendix B.2 for details.

For the total daily hospital admissions data and the care home deaths data we retain the negative binomial model, with the respective $\kappa$ parameters free to be estimated. Some overdispersion here is a pragmatic way to deal with likely model mismatches in these components. For example, in addition to the mismatches expected from not modelling hospital acquired infections (e.g. [13]), it seems likely that there was some on the ground variability in the severity of disease sufficient for hospitalization, and in rates of discharge, particularly early in the epidemic and when loads were high. For the hospital deaths we set $\kappa = 2000$, which gives a likelihood very close to Poisson. There is no legitimate reason to expect overdispersion here, if the model is at all fit for purpose.

## Relaxing the model assumptions

The largest change made here is to relax the strong assumption that $b(t)$—which represents the effects of NPIs, *the weather and other factors*—is a piecewise linear function with slope changes only at 12 selected NPI change points. Here, $b(t)$ is instead represented semi-parametrically by a logistic transform (see S1 Appendix B.1) of an adaptive smoothing spline, with 80 coefficients and 5 smoothing parameters, in which the degree of smoothness is allowed to vary smoothly

with *t*. See section 5.3.5 of [24] for details. The point of this change is to use a representation of *b(t)* that allows for a *much* wider range of possible function shapes and a well founded data driven means for choosing between them, thereby greatly increasing the role of the data in the estimation of *b(t)*, while reducing the role of prior assumptions. Of course it does nothing to remove the confounding of NPIs with weather and other effects, such as spontaneous behavioural changes, but it does avoid the implication that the weather and people's behaviour change their course only in response to government announcements.

We also relaxed the assumption that all the $\gamma$ parameters are fixed and known. Firstly, the reference used to justify the choice of $\gamma_G$, controlling the rate of progression of fatal disease in care homes [25], appears to contain no information on this parameter, so we allowed it to be a free parameter, which slightly reduces care home death mistiming. Secondly, the model also has difficulty matching the general ward and ICU occupancy data, tending to over-estimate both in the Midlands and two northern regions. To reduce this problem it seemed reasonable to relax the assumption that all the rate parameters controlling progression through the health system were fixed and known. In particular we relaxed the parameters for which there seemed likely to be most scope for some latitude in clinical judgement, perhaps driven by local circumstances, to make substantial differences. So we relaxed the assumptions on the rates related to movement of recovering patients through the system. That is $\gamma_{IC_{W_r}}$, $\gamma_{W_r}$ and $\gamma_{H_r}$ were treated as free parameters.

A final rigidity in the model structure is that there is assumed to be no infection before individuals could at least potentially become symptomatic on leaving the *E* stage. At the same time the mean duration of the symptomatic infective stage is set equal to the mean time from symptom onset to hospitalisation. This makes for a very long generation time, much longer than the 5–7 days reported in the literature for the serial interval or generation time (see p. 26 of [9] for a review). One consequence of this is that *R* estimates need to be higher than those usually quoted to meet the initial rate of increase in the disease ([1] actually limits *R* in a way that avoids estimates being too high). To relax this link between clinical disease progression rates and the generation interval, we introduced an extra compartment between $I_c$ and hospitalization (see the grey 'P' on Fig 2).

$$\dot{P} = \gamma_C I_c - \gamma_{ph} P$$

where *P* replaces $I_c$ in all flows into hospital compartments and the *R* state. By appropriate choice of $\gamma_{ph}$, this state allows us to shorten the *E* state and $I_c$ state, hence reducing the generation time, without changing the literature based mean time from infection to hospitalisation. Specifically, we shortened the *E* state to have an average of 3 days to infectivity, and the $I_C$ state to be 4 days, yielding a generation time of 6.2 days (accounting for the duration of $I_A$, which was unchanged). The *P* state then has an average duration of 5.3 days so that the total time from infection to hospitalization still matches the literature based 5.8 + 6.5 days discussed previously.

### Estimation and inference

The sensitivities of the model states with respect to the parameters were obtained for all 703 model state variables, yielding a system of 65379 sensitivity ODEs. Model and sensitivities were solved by fourth order Runge-Kutta integration (see e.g. [26]) with a one day time step (having confirmed that halving the step made negligible difference to the evaluated likelihood). Hence the log likelihood and its derivatives w.r.t. the free parameters could be readily evaluated. Due to sparsity and cache efficiency, the sensitivity system less than doubles computing time for the model. Computing the likelihood, likelihood derivatives and *R* series for the full

model takes less than a second on a single core of a low specification laptop—it is considerably faster for the original [1] model with fewer free parameters.

Given the log likelihood and derivatives, the penalized log likelihood and derivatives are also readily evaluated, so the posterior modes of the free parameters can be obtained by quasi-Newton optimization. The smoothness of $b(t)$ was controlled by a Gaussian smoothing prior, with 5 free smoothing parameters, which were estimated by the approximate marginal likelihood optimization method of [27]. Uncertainty was assessed using the large sample approximate posterior covariance matrix of the parameters, and the delta method. See S1 Appendix B.3.

## Results

Fig 3 shows the fit of the model with the various assumption relaxations applied. The model fits imperfectly, with some systematic errors in the fit to hospital occupancy and arrival data as expected: without modelling the hospital acquired infections (which are included in the data and, as discussed previously, often made up a substantial portion of the total hospitalized), as well as possible time variability in on-the-ground admission criteria, it is unlikely that better fits could be achieved. Given the ambitious nature of the fitting task, it seems reasonable to view the results as useful in the statistician George Box's 'all models are wrong, but some are useful' sense.

Figs 4 and 5 show the corresponding inferences about incidence and *R*. All regions have peak incidence prior to the first lockdown with total incidence for England in decline well before lockdown. The regional incidence picture is more mixed at the second lockdown, although the total is again falling well before lockdown. Furthermore all regions have $R \lesssim 1$ by either lockdown, with average $R < 1$ some days before either lockdown. Several regions relatively distant from London have the inferred *R* initially increasing. This is probably an artefact caused by the independent initialisation of each region, which cannot capture the initial region-to-region spread. As in [1] the plotted uncertainties would be over-optimistic, even if we assumed a correct model structure, as they do not account for the uncertainty in most of the rate constants.

Although it could also be partially weather driven, the systematic pattern of *R* continuing to fall after the first lockdown is introduced, and then increasing again well before the lockdown restrictions were lifted, is to be expected. *R* is the average number of new infections *per existing infection*. Immediately after lockdown most infections are in the locked down population, with a low *R*, and only a minority are in the key worker population with higher *R* (assuming lockdown has an effect), so the average is low. After the locked down population runs out of household members to infect, the proportion of infections among key workers must increase, due to their higher *R*. So the average *R* must increase too as most of the infections to average over are now in the higher *R* population. Although the simple arithmetic mechanism underlying this effect results from having locked down and key worker strata, we only observe aggregate data, reflecting the change in *R*, but not what causes it. The model also deals only with populations aggregated over the two strata, but can still capture the change in *R* apparent in aggregate data, if $b(t)$ is flexible enough. However, the piecewise linear $b(t)$ of [1] is not flexible enough in this regard.

Fig 6 shows how the lockdown 1 timing result depends on the various changes made to the [1] model, when they are applied sequentially. All panels use the corrected likelihood. The top left panel then uses the incubation period and time to hospitalization used by [1], and the same serial interval, but has the piecewise linear $b(t)$ replaced by an adaptive spline. Rather than *R* being much larger than 1 on the eve of lockdown it is around 1. The top right panel modifies the model further, by reducing the serial interval to about 6.2, making it closer to the literature range—if anything this moves the $R = 1$ point slightly later. The bottom left panel is

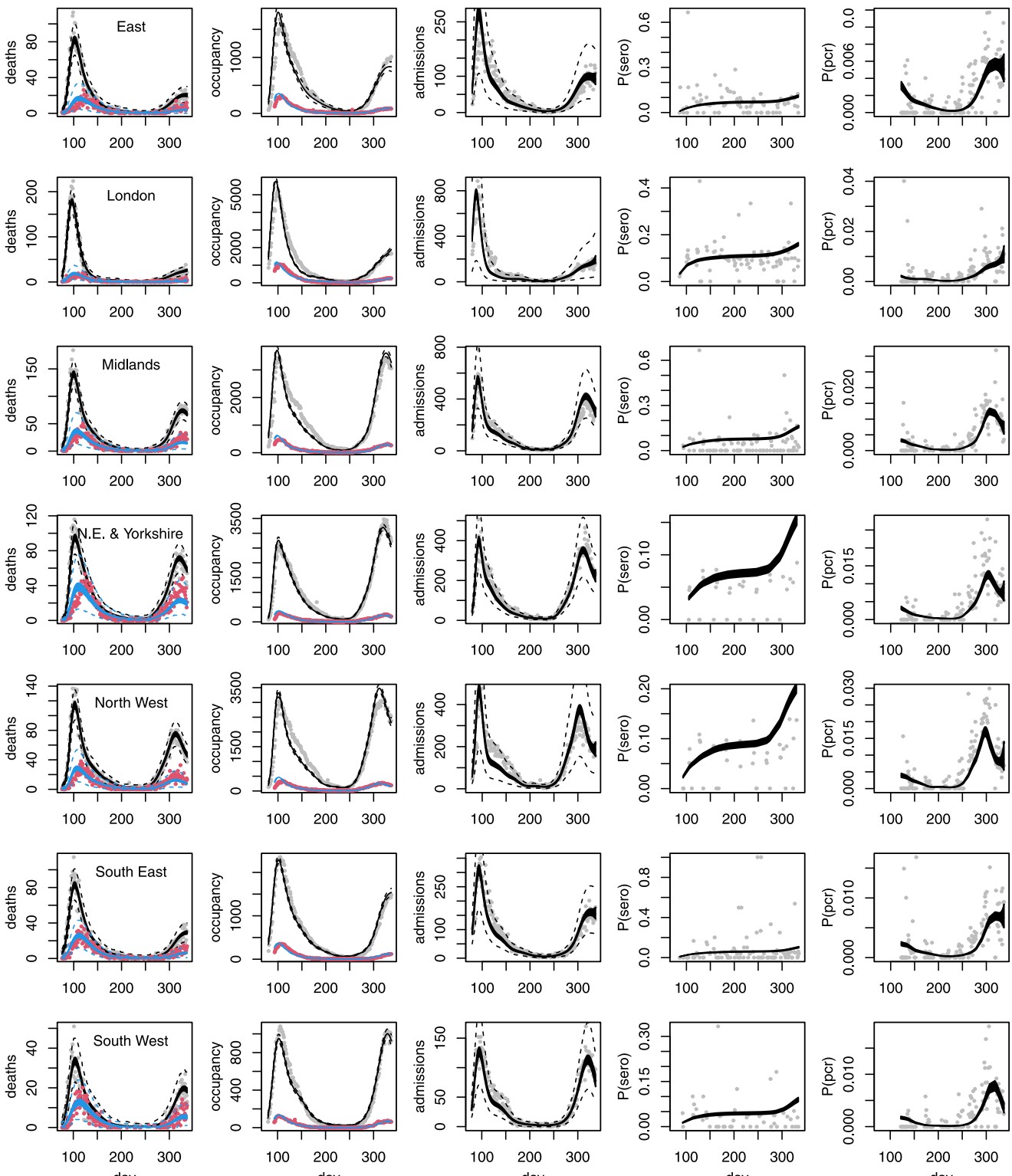

**Fig 3. Model fits (posterior 95% credible bands for expectations) to the death, hospital and testing data, with one region per row, against day of year.**
In the leftmost 'deaths' column, grey points are hospital deaths and red points care home deaths. In the second 'occupancy' column, grey is general ward occupancy and red ICU occupancy. For the deaths and hospital admissions 95% prediction interval limits are shown as dashed curves. Prediction intervals are not reported for occupancy, where the likelihood is based on differencing, or for the test data, where highly variable sample sizes gives intervals showing no statistical problems, but which are visually unpleasant. Note some substantial discrepancies in the two northern regions.

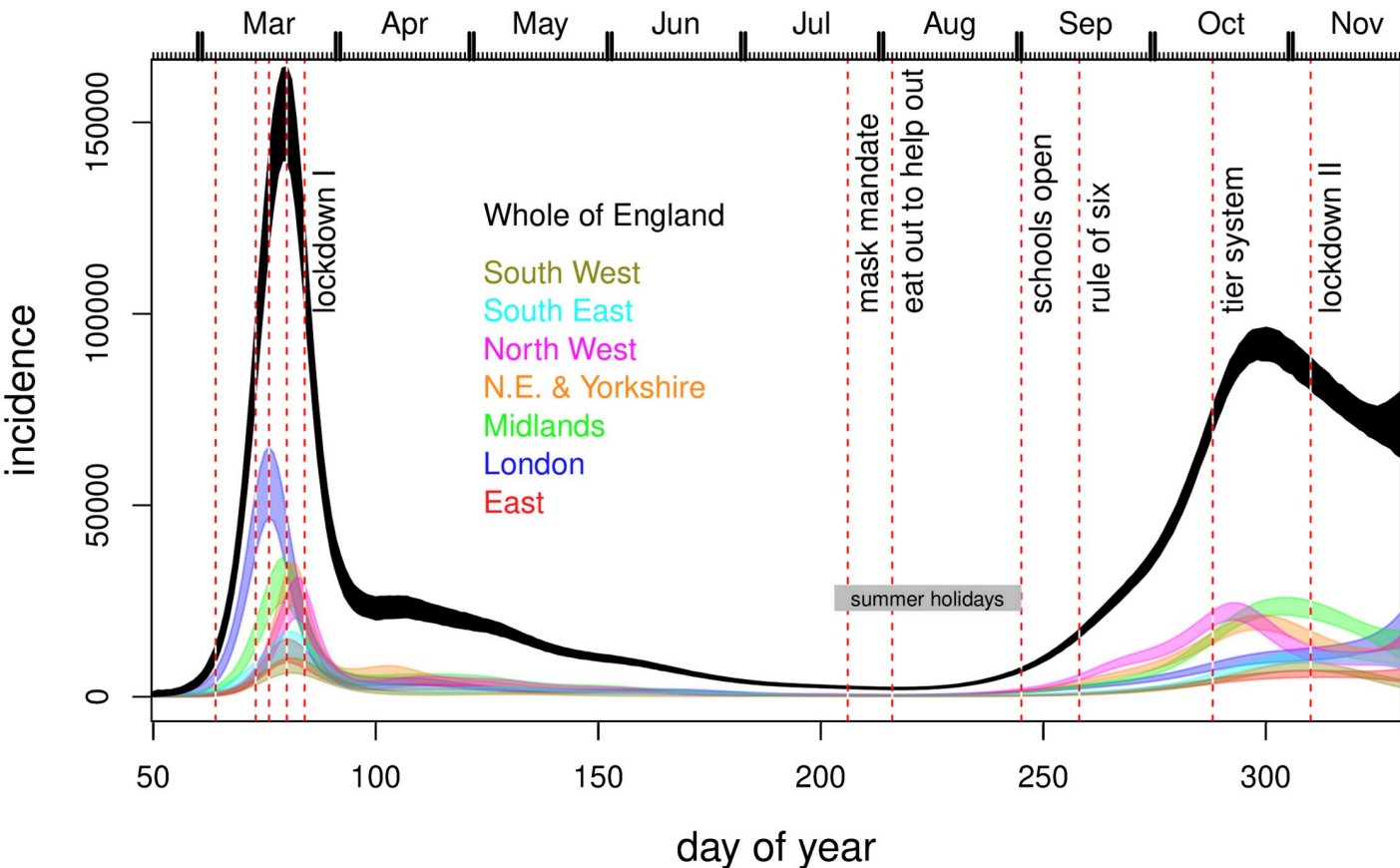

**Fig 4. Inferred incidence, for all regions (coloured) and whole of England (black).** Notional 95% credible bands are shown. These do not reflect all the uncertainty in rate parameters and assume a correct model structure: hence they provide a lower bound on uncertainty. Vertical dashed lines show some policy changes. The 4 preceding lockdown I are information campaign, symptomatic self isolation, work from home advice, school and hospitality closures. 'Eat out to help out' was a scheme encouraging people to use the restaurants and pubs. The re-opening of schools after the first lockdown is also shown. Subsequent policies introduce increasing levels of restriction.

then the model with the incubation period and time to hospitalization set to the literature values consistent with the papers cited in [1] as the sources of these durations. This panel is simply an enlargement of the relevant portion of Fig 5. Finally the lower right panel shows the results when the smoothing penalty is downweighted by a factor of 4. This checks whether the timing results could be driven by smoothness assumptions, by substantially reducing the amount of smoothing relative to the estimated level. The results do not appear to be a smoothing driven artefact.

If fits of this model to data are viewed as a reasonable basis for inference about the timing of incidence and *R* levels, then the implication is that *R* < 1 probably occurred some time before both the first two English lockdowns, and that incidence was already in sharp decline before either. The contrary result of [1] relies on a very restrictive model for *b*(*t*) and on setting incubation and hospitalization times to values less than those given in the papers cited as their source.

## Discussion

Three major claims are made in [1]. Whereas the first is of a descriptive nature, namely that the two English Covid-19 lockdowns in March and November 2020 coincide with a major

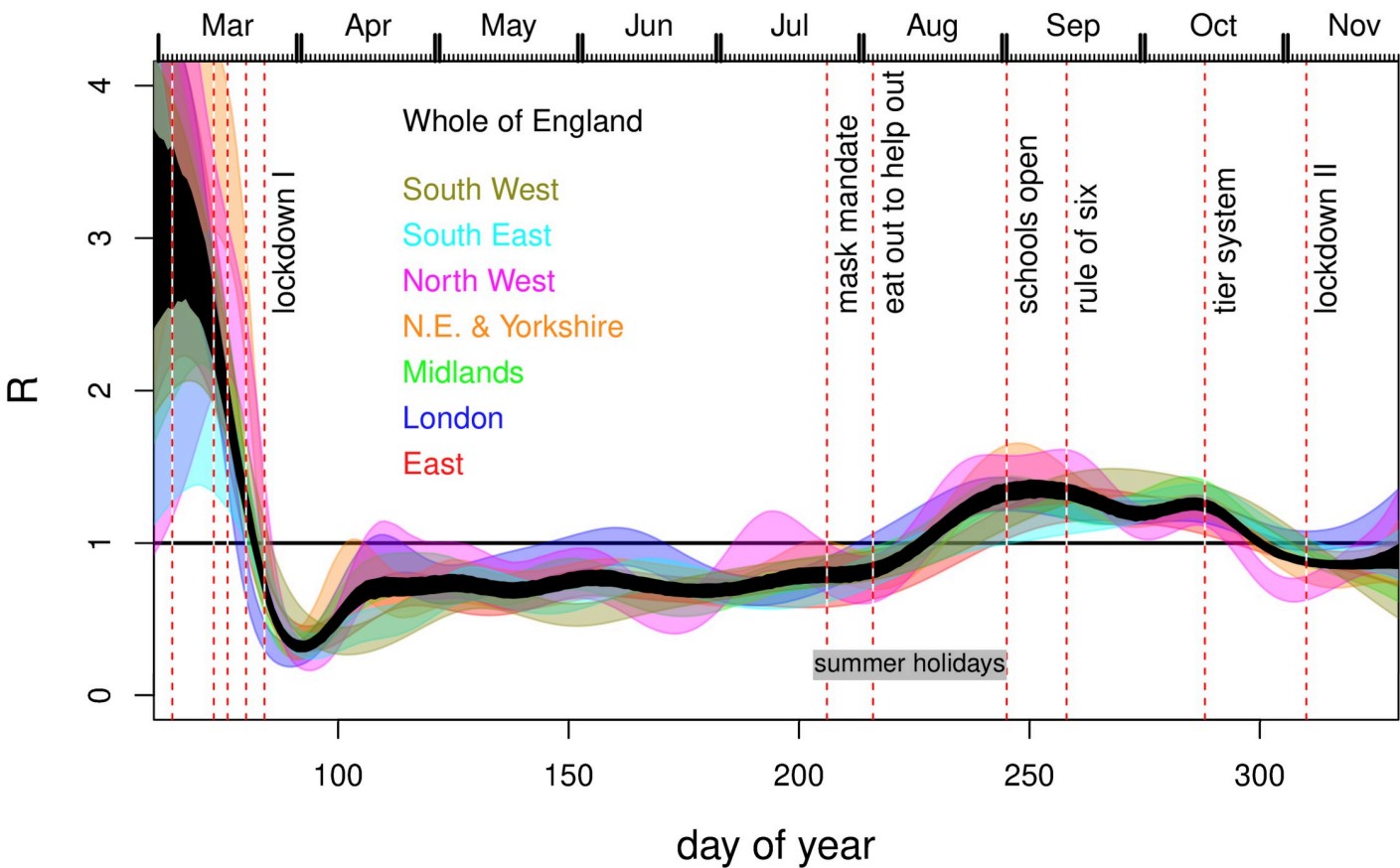

**Fig 5. Inferred *R* for all regions (colour) and the infectives-weighted average for the whole of England (black).** Notional 95% credible bands are shown. These do not reflect all the uncertainty in rate parameters and assume a correct model structure: hence they provide a lower bound on uncertainty. Vertical dashed lines as Fig 4.

drop in the reproduction rate of Covid-19 in the UK, the other two are of a so-called "counterfactual" nature: (i) if England had not gone into lockdown, then there would have not been an associated drop in reproduction rate and (ii) if England had gone into lockdown earlier (or later) then a lot of lives would have been saved (or lost, respectively).

The key challenge is that a counterfactual cannot be directly observed and must be approximated with reference to a comparison group. There are various accepted approaches to determining an appropriate comparison group for counterfactual analysis, ideally using a prospective design. When this is not available, such as in this case, a retrospective approach is necessary. But there are stringent conditions on a retrospective design in order for it to have counterfactual validity, such as avoiding confounding, contamination, and impact heterogeneity (see [6] for an introductory treatment). Confounding occurs where certain factors, for example the various social distancing measures in place prior to the lockdowns, are correlated with exposure to the intervention and, independent of exposure, are causally related to the outcome of interest. Confounding factors are therefore alternate explanations for an observed, but possibly spurious, relationship between intervention and the outcome; in this case between lockdown and the reduction in *R*. The pre-lockdown social distancing measures are also an example of contamination, which may also invalidate any counter-factual statements. Contamination occurs when members of treatment group (i.e. the actual population) and/or comparison groups (i.e. the counterfactual populations) have access to another intervention which also

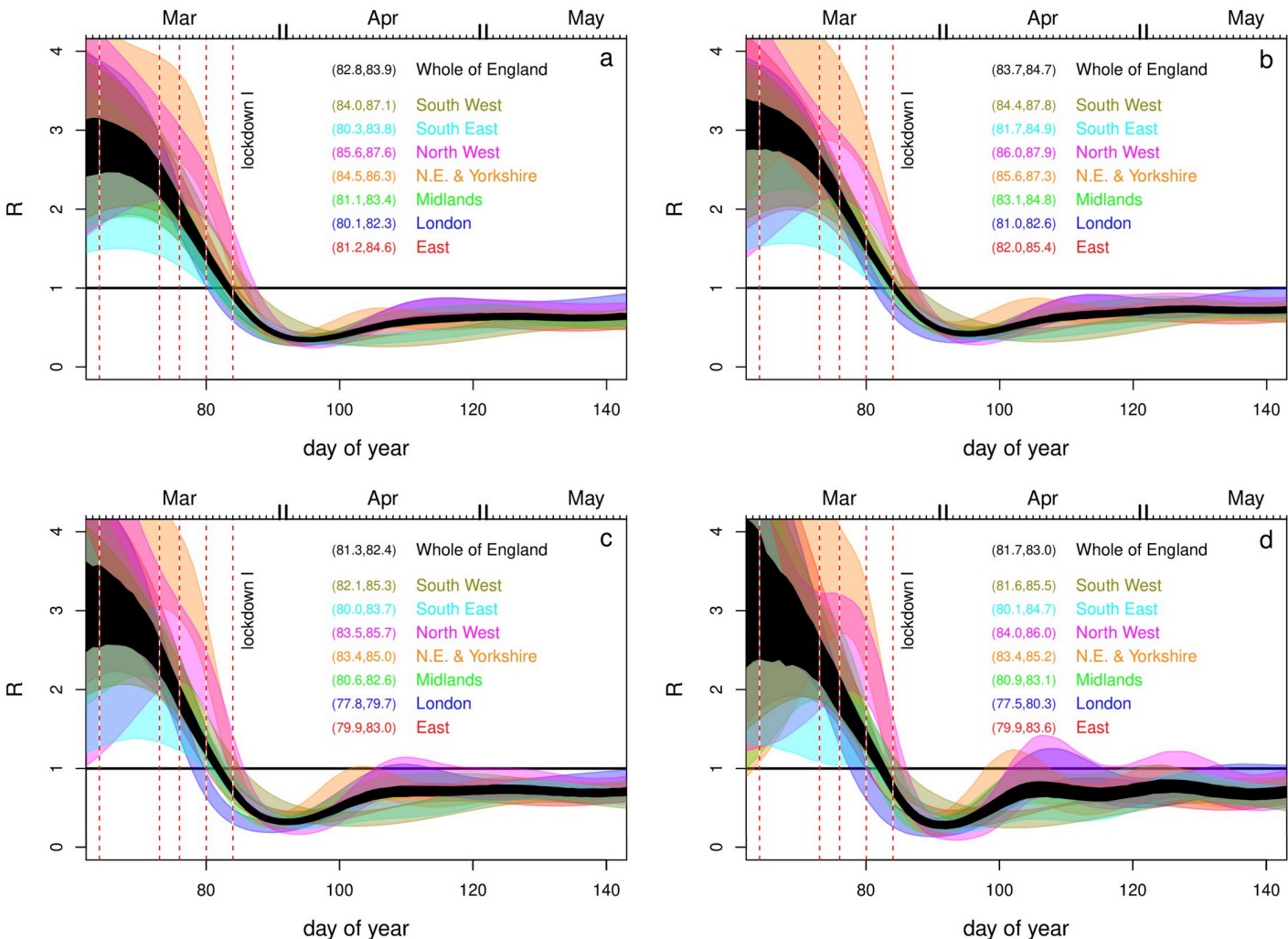

**Fig 6. Comparison of inference around Lockdown I (March 24th 2020, day 84), for different modifications of the model of [1]. a.** Piecewise linear *b*(*t*) replaced by adaptive spline. **b.** As a, but adjusting serial interval down towards literature range. **c.** The fully corrected model. As b, but with incubation period and time to hospitalisation set to values given in the papers cited by [1] for these quantities. **d.** As c, but with the smoothing parameters reduced by a factor of 4 from their estimated values as a sensitivity check. The numeric intervals given are nominal 95% CIs for the day on which *R* < 1 first occurred.

affects the outcome of interest. Additionally, there is the issue of impact heterogeneity: the impact of the lockdown will be very different in the locked down subset of the population, compared to key workers, who are less restricted. Finally, [1] explicitly states that *b*(*t*) is modelling both the effects of NPIs and the weather. There is therefore no basis on which the model can identify the effect of lockdown independent of the weather, enabling the counterfactual manipulation of one while appropriately controlling the other. But such control is absolutely fundamental to causal reasoning with counterfactuals. We conclude that the model and inference of [1] do not form a reasonable basis for making counterfactual statements about how many people would have died if lockdown had occurred at a different time. Even without the preceding general problems, there is the specific problem that lockdown can not have caused *R* to drop below one if this event preceded lockdown, but the counterfactual statements rely on such a causal link.

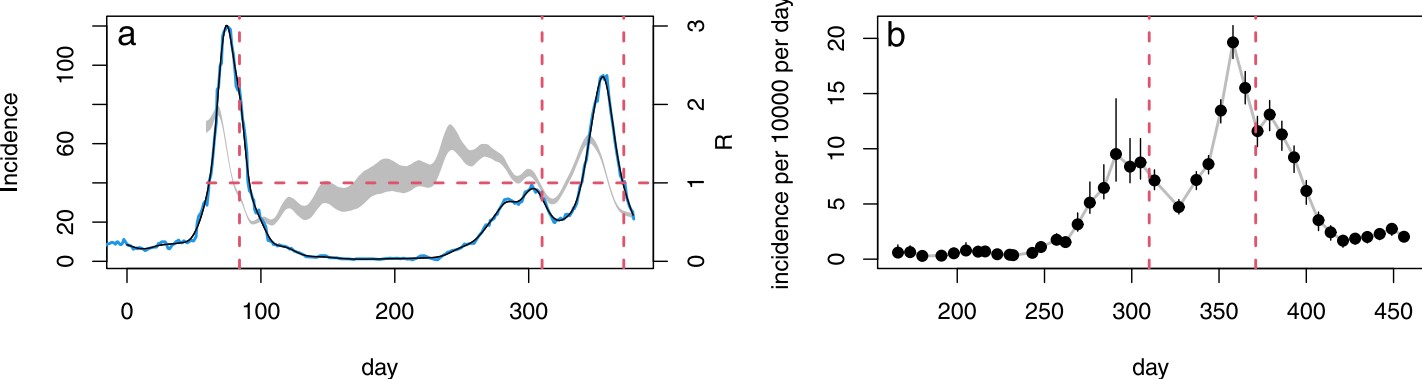

**Fig 7.** **a**. continuous curves are onset of new symptoms per day from the REACT-2 study digitized from [29], and lagged by the average 5.8 days from infection to first symptoms to give incidence: blue is raw and black is spline smoothed. Jan 1 2020 is day 1 and vertical dashed red lines show the lockdown dates. The grey band shows a 95% credible interval for *R* reconstructed from the smoothed incidence curve by the method given in section 5.1 of [28]. The horizontal dashed line shows *R* = 1. Incidence peaks about 9 days prior to lockdown 1 (day 84, March 24th 2020), and *R* < 1 four days before lockdown 1. **b**. [30] published estimated incidence with 95% confidence limits. Red lines show the dates of the second and third UK lockdowns—the survey was not running at the first.

While this paper was in review, more direct evidence emerged which aligns with our conclusions, but not with [1]. [28] used a direct statistical deconvolution approach to infer incidence from hospital death data and three published infection to death distributions. The study gives similar results for incidence and *R* to the whole England results obtained here, and its conclusions are strengthened by the close match between the disease duration distributions used and more recent disease duration data reported by [22] based on more than 24,000 fatal cases. The results here and in [28] also correspond to the reconstructions of the number of newly symptomatic infections each day, reported by [29]. This latter study is based on symptom onset dates reported by antibody positive subjects in a properly randomized surveillance sample. Lagged by the average latent period this gives a direct estimate of incidence, and the results are shown in the left panel of Fig 7. The incidence reconstruction can also be used to infer *R* by the method given in section 5.1 of [28], and this reconstruction is also shown. Finally, the UK Office for National Statistics has published incidence estimates based on its properly randomized Covid-19 surveillance survey. The survey was not yet active at the time of the first peak, but its results (see Fig 7, right) are in agreement with [28, 29] and the results reported here for the second half of 2020. Hence our model fitting based results are consistent with the relatively direct estimates based on the three least biased data sources available.

After we had received referees reports for this paper (on 18th June 2021), and revised accordingly, [1] was published in *Science Translational Medicine* [31], having been submitted there on 14th April 2021. The published paper does not refer to our work, but made some changes relative to [1], of which the most significant appear to be: (i) introducing a pre-hospital non-infectious stage, equivalent to our 'P' stage, to shorten the generation time/serial interval to be consistent with the literature and (ii) estimating two common negative binomial *κ* parameters, thereby avoiding simply setting them to 2 (the number of particles used in filtering has been increased accordingly). An extra 'community deaths outside hospital' data stream (comparatively small numbers) was also fitted. The main results of [31] are essentially the same as [1], although the new equivalent of Fig 1 now shows London as having *R* < 1 before the first lockdown, and *R* for other regions is slightly reduced on the eve of lockdown. Significantly, given our results, the *b*(*t*) model was unchanged and the time to hospitalization, incubation time and hospital occupancy likelihoods remain uncorrected in [31]. No modification appears to have been made that might enhance the statistical validity of the 'counterfactuals'

presented. Hence we do not believe that the changes made between [1] and [31] address the most substantial issues raised here or undermine our results.

Our results on the timing of $R < 1$ and peak incidence obviously do not imply that the lockdowns had no effect. Indeed the dip and recovery seen in $R$ after the first lockdown is *only* expected if lockdown reduces spread in the locked down population, relative to those not locked down. The point is rather that the additional effect, on top of the cumulative effects of other behavioural changes pre-dating lockdown, seems likely to have been greatly overstated. In our view, determining definitively what caused $R$ to drop below one is not possible. In March especially, policy and behavioural changes were so rapid (public information campaign, March 4th; symptomatic self isolation 13th; work from home advice, 16th; school and hospitality closures 20th; full lockdown, 24th) that there would simply have been insufficient time to determine what had worked, even if adequate data had been gathered to answer this question. In fact, there was no surveillance testing at that point. However, it seems difficult to make the case that full lockdowns were necessary to bring $R$ below one, whether region-by-region or in aggregate for England. In densely populated London, by far the UK's largest city where the control problem should be most difficult, the evidence is particularly strong that $R < 1$ well before full lockdown. While not impossible, it would be quite counter-intuitive if stronger measures were in fact necessary for control in the less densely populated regions.

## Supporting information

**S1 Appendix. Supplementary appendices.**
(PDF)

**S1 Code. Replication code and data.**
(ZIP)

## Acknowledgments

We thank the 2 referees and the editor for some helpful suggestions for improving the paper, including the suggestion of Fig 6. Thanks also to Nicole Augustin, Fraser Nelson, Jason Matthiopoulos and Jonathan Rougier for useful comments and discussions. We supplied the preprint version of this paper to the authors of [1] on 4th February 2021, when we posted a copy on medArxiv. They acknowledged receipt, but have not responded further.

## Author Contributions

**Conceptualization:** Simon N. Wood, Ernst C. Wit.

**Formal analysis:** Simon N. Wood, Ernst C. Wit.

**Investigation:** Simon N. Wood, Ernst C. Wit.

**Methodology:** Simon N. Wood.

**Project administration:** Simon N. Wood.

**Software:** Simon N. Wood.

**Validation:** Simon N. Wood, Ernst C. Wit.

**Visualization:** Simon N. Wood.

**Writing – original draft:** Simon N. Wood, Ernst C. Wit.

**Writing – review & editing:** Simon N. Wood, Ernst C. Wit.

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
