## [Decision Letter · Decision Letter 0]

18 Jun 2021

PONE-D-21-08472

Was R < 1 before the English lockdowns? On modelling mechanistic detail, causality and inference about Covid-19

PLOS ONE

Dear Dr. Wood,

Thank you for submitting your manuscript to PLOS ONE. After careful consideration, we feel that it has merit but does not fully meet PLOS ONE’s publication criteria as it currently stands. Therefore, we invite you to submit a revised version of the manuscript that addresses the points raised during the review process.

Both reviewers provided constructive and detailed suggestions to improve the manuscript quality and presentation, especially considering that the manuscript provides a criticism of results obtained by another research group [Knock et al., 2020]

I echo one of the reviewers in the suggestion of putting the authors claim in a more objective fashion, in order to provide the readers with a balanced counterpoint to [Knock et al., 2020], rather than a mere criticism of their work, especially in light of the fact that both studies make simplifying assumptions that might bias their claims. 

In this view, another reviewer requires more discussion about the authors' assumptions and the criticism of [Knock et al., 2020] ones. I agree with the reviewer that a more detailed and balanced discussion is required. In my opinion, as well as according to many other scholars, it is very hard to disentangle the single effects of different measures on the spreading of COVID-19, and such a study is strongly dependent on the simplifying assumptions made to carry out the study. Hence, I think that the proposed manuscript would greatly benefit by the analysis of a more balanced viewpoint, highlighting strengths and weaknesses of both set of assumptions, and, possibly, to the consequences that such assumptions may entail in both studies. 

We look forward to receiving your revised manuscript.

Kind regards,

Alessandro Rizzo

Academic Editor

PLOS ONE

Journal Requirements:

1. Please ensure that your manuscript meets PLOS ONE's style requirements, including those for file naming. The PLOS ONE style templates can be found athttps://journals.plos.org/plosone/s/file?id=wjVg/PLOSOne_formatting_sample_main_body.pdf and https://journals.plos.org/plosone/s/file?id=ba62/PLOSOne_formatting_sample_title_authors_affiliations.pdf

2. Please include a copy of Table 1 which you refer to in your text.

Additional Editor Comments (if provided):

Reviewers' comments:

Reviewer's Responses to Questions

**Comments to the Author**

1. Is the manuscript technically sound, and do the data support the conclusions?

Reviewer #1: Partly

Reviewer #2: Partly

2. Has the statistical analysis been performed appropriately and rigorously? 

Reviewer #1: Yes

Reviewer #2: Yes

3. Have the authors made all data underlying the findings in their manuscript fully available?

Reviewer #1: Yes

Reviewer #2: Yes

4. Is the manuscript presented in an intelligible fashion and written in standard English?

Reviewer #1: Yes

Reviewer #2: Yes

5. Review Comments to the Author

Reviewer #1: The manuscript contains a sophisticated compartmental model which allows for capturing the course of COVID-19 pandemic in England. In particular, by fitting the model to different epidemiological indicators such as the number of fatalities, hospitalizations or the evolution of test positivity, the authors are able to infer the actual epidemic incidence, i.e. the number of daily new cases, and study the evolution of the effective reproduction number, which is a pivotal indicator to quantify the stage of an ongoing outbreak.

The analysis presented by the authors represents a critical assessment of a report published by an independent research group in December 2020 addressing the same problem [Knock et al. 2020]. There, it is claimed that lockdown measures during the first epidemic waves in England were essential to reduce the effective reproduction number below one and, as a consequence, observe a decrease in the daily incidence.

The findings here contradict the claims made in [Knock et al. 2020] and show that the epidemic incidence started decreasing before full lockdown measures were implemented in England. The discrepancies between both results mainly arise from three factors: i) the choice of a shorter generation time interval, ii) the correction of the typical time elapsed from the onset of symptoms to hospitalization and iii) the continuity of the function reflecting the impact of non-pharmaceutical interventions. The mathematical framework proposed here is solid and described in depth and the methodology followed by the authors is clearly explained, which make the manuscript scientifically sound.

My main criticism with the current manuscript concerns its writing style. In this sense,I think that the authors should tone down some parts of the manuscript to make their critical assessment more objective. I would also encourage the authors to keep their analysis at the scientific fundamental level rather than discussing further political implications. Some examples of parts which, in my opinion, should be rewritten are:

Page 2: The comments about the media attention of the reports published by the team from Imperial College London or their relevance for policy-making lie out of the scope of a scientific assessment of the model.

Page 3: “Knock et al. do not report such checks, showing only the outputs of filtering” I think that this does not provide any valuable information for the discussion made in the paragraph.

Pages 4-5: “The model captures many features in impressive detail, but several aspects are not modeled”. The enumeration following this sentence describes very exhaustively many limitations of the model introduced in [Knock et al. 2020], most of them are not introduced in the model proposed here by the authors either. As in any theoretical model, some assumptions should be made to reduce the number of equations and not to end up with an unmanageable huge parameters’ space. As the submitted manuscript does not incorporate most of these logical limitations of theoretical frameworks, I do not see the point of including such exhaustive description.

Page 6: “It is hard to understand this choice, unless it was made to avoid particle depletion problems in filtering”. This statement seems very subjective and can be expressed in other words or left to the interpretation of the reader.

Page 7: “While not ideal, this is a less wrong assumption” Given the fact that the assumption is not supported by any reference either, although I agree it can be more logical, I would avoid making any comparison.

Page 9: “Given the ambitious nature of the fitting task, it seems reasonable to view the results as useful in the ‘all models are wrong, but some are useful’ sense.”

Page 12: “We conclude that the model and inference of Knock et al. do not form any sort of reasonable basis for making counterfactual statements about how many people would have died if lockdown had occurred at a different time”

Concerning the contents of the manuscript, I have some suggestions which, in my opinion, would make the result presented by the authors more robust:

Page 2: For the sake of readability, it would be very useful to include some references supporting how lockdowns are detrimental to the evolution of the pathogen or explaining the economic side effects observed in many countries.

The expression for the time derivative of the sensitivities of each variable with respect to the different parameters is valid when the parameters $\\theta$ are constant and do not depend on time. Nonetheless, the function $b(t)$ is time-dependent and, if I am correct, this expression should not be valid in this case. Please clarify it.

I think the manuscript would clearly benefit from a figure showing explicitly how the evolution of the effective reproduction number depends on the generation time interval chosen or the time to hospitalization as reflected by the authors in Page 9.

I would avoid mentioning seasonality as a crucial confounding factor invalidating the counterfactual scenarios proposed in [Knock et al. 2020] because, in the short time windows analyzed in the latter report, one cannot expect substantial underlying changes in the weather conditions.

The section describing the computation of the effective reproduction number should be extended to better explain the significance of this indicator. In particular, within the different alternatives explored in the literature (see [Gostic et al. PLoS Comput Biol 16(12): e1008409 (2020)] for a further review on the topic), the authors compute the reproduction number as the expected number of contagions that an agent becoming infectious at time $t$ will make throughout his/her infectious period if the conditions of the system remain immutable.

In this sense, the explanation of the dependence of the effective reproduction number after lifting lockdown measures in page 9 is not valid if one resorts to this definition. Providing the model does not divide the population into key workers and locked-down individuals either, I think that the authors could get rid of this discussion.

Finally, to obtain the reproduction number at the national level, the authors should weight the individual reproduction numbers by the newly infected individuals in each of the regions rather than with the infected population.

Reviewer #2: The manuscripts provides a revision of the study by Knock et al. (2020). The authors highlight various caveats and adapt the model accordingly. The main finding is that, in contrast to Knock et al., the reproduction number is found below one before the implementation of the two lockdowns in England. In this sense, the authors then question whether the lockdowns were the main driver to control SARS-CoV-2 in England.

The approach the authors take is well motivated and scientifically sound. The authors highlight various shortcomings of the study by Knock et al. and adapt different aspects of the model. Unfortunately, due to the various changes made, it is not possible to isolate the aspect that lead to the different result. It may be due to the shorter generation time, the adapted time from symptom onset to hospitalisation or the different functional form of R(t). I guess that the fit is very costly to perform, but I strongly recommend to incrementally adapt the model by Knock et al., which would allow to isolate the impact of the various changes made.

The inferred reproduction number R(t) starts to decrease almost immediately (Figure 5). How do the authors explain this early decrease? I guess no NPIs were in place back then and mobility still showed no reduction. In this sense, please explicit in a table when the reproduction number actually was below one. It seems that during the first wave the reproduction number in London was below one surprisingly early. Additionally, I strongly recommend the authors to work with dates instead of integer numbers for the day of the year. It would improve the readability of the manuscript a lot.

In the manuscript here, the model consists of regular ODEs, whereas Knock et al. worked with a stochastic model. The authors write: “The neglect of stochasticity in the state equations seems likely to be a minor issue here, relative to the other approximations made in the model.” In principle I agree with this statement. However, looking at the results (Figure 3) the peak in care home deaths is anticipated with respect to the data in all regions. One explanation could be the lack of stochasticity that may delay infections in care homes due to the modular contact matrix. Another factor could be the different initialisation the authors choose as it is pointed out in the manuscript.

In Figure 3, one can see that occupancy and admissions generally peak earlier than the data and decrease faster. How do the authors explain this mismatch? Furthermore, why is there no uncertainty range indicated in Figure 3? Is it too small to be visible? However, this would not be consistent with the uncertainty ranges presented for the incidence (Figure 4) and the reproduction number (Figure 5).

In the discussion, the authors mention a sensitivity analysis they performed. In particular, they considered different time to hospitalisation, generation time and likelihood for occupancy. Please show these results in a supplementary. Because comments like “… R = 1 a little later, but still before lockdown” are not very helpful for the reader. Indicate how much later and when if you already performed the fit.

The authors criticise many times that Knock et al. did not isolate the impact of seasonality and NPIs. Furthermore, you claim that this invalidates the counterfactual scenarios. If the counterfactual scenarios consisted of shifting the time series by months this would certainly be true. However, Knock et al. consider a maximal shift of two weeks. Do the authors truly believe that the meteorological change in two weeks invalidates the conclusions of Knock et al.?

The authors criticise that Knock et al. did not model within hospital transmission. Furthermore, they claim within hospital transmission account for a quarter fo cases in both waves and cite a health service journal. Is this with respect to the reported cases or do the authors believe this also holds for infections in general? Be more specific whether you refer to reported cases, which will be strongly biased towards within hospital transmission. Furthermore, making such claims based on a reference to a journal does not seem appropriate for a scientific article.

The authors point out that including data regarding the number of tests undermines the analysis and should therefore not be included. However, I do not understand whether you actually did fit testing data or not. Later in the manuscript you explain the different assumption made regarding the pre-tested rate. This led me believe that you actually fit to the testing data. Please clarify this point. Also you should be consistent, if you truly believe that it undermines the analysis then don’t include it.

A more general remark is regarding the writing of the manuscript. Various parts are not written with the objectivity a scientific article requires. For example, criticising the approach of Knock et al. for not modelling the differences between men and women seems really far fetched. Similarly, on page 6, you refer to a reasonable model. Rewrite this such that it is less judgemental. An example would be “a more accurate/better approach”. The same comment also holds for other parts of the manuscript:

Page 7: “Less wrong assumption … “. Change it to better/ more realistic assumption.

Page 2: “… was accompanied by substantial press release material”. Why is this relevant?

Page 1: You insinuate that the lockdown lead to a substantial loss of life due to the caused economic hardship. A scientific article is not the place to make insinuations. Referencing a pre-COVID study to back up a claim about the impact of lockdown is very vague and implicit. If you want to make such comment find a study that actually treats the impact of lockdowns and aims at quantifying the non COVID related loss of life.

Page 12: You write “We conclude that the model and inference of Knock et al. do not form any sort of reasonable basis …”. I understand your criticism of their approach, but saying that it does not support any reasonable basis is an exaggeration.

6. PLOS authors have the option to publish the peer review history of their article (what does this mean?). If published, this will include your full peer review and any attached files.

Reviewer #1: No

Reviewer #2: **Yes: **Benjamin Steinegger

---

## [Author Response · Author response to Decision Letter 0]

1 Jul 2021

Please see the attached Response to reviewers file

---

## [Decision Letter · Decision Letter 1]

5 Aug 2021

PONE-D-21-08472R1

Was R < 1 before the English lockdowns? On modelling mechanistic detail, causality and inference about Covid-19

PLOS ONE

Dear Dr. Wood,

Thank you for submitting your manuscript to PLOS ONE. After careful consideration, we feel that it has merit but does not fully meet PLOS ONE’s publication criteria as it currently stands. Therefore, we invite you to submit a revised version of the manuscript that addresses the points raised during the review process.

The two Reviewers are appreciative of the work done on the manuscript, which has now largely improved. Both reviewers raise some minor points toward making the paper even clearer and impactful. In particular, one of the Reviewers requires some more discussion about the definition and implications of lockdowns and/or other restrictions, whereas the second one requires some clarifications about the use of the reproductive number and suggests to remove the section devoted to the correspondence between the authors of this manuscript and the team of Knock's et al. 

We look forward to receiving your revised manuscript.

Kind regards,

Alessandro Rizzo

Academic Editor

PLOS ONE

Journal Requirements:

Reviewers' comments:

Reviewer's Responses to Questions

**Comments to the Author**

1. If the authors have adequately addressed your comments raised in a previous round of review and you feel that this manuscript is now acceptable for publication, you may indicate that here to bypass the “Comments to the Author” section, enter your conflict of interest statement in the “Confidential to Editor” section, and submit your "Accept" recommendation.

Reviewer #1: (No Response)

Reviewer #2: (No Response)

2. Is the manuscript technically sound, and do the data support the conclusions?

Reviewer #1: Yes

Reviewer #2: Partly

3. Has the statistical analysis been performed appropriately and rigorously? 

Reviewer #1: Yes

Reviewer #2: Yes

4. Have the authors made all data underlying the findings in their manuscript fully available?

Reviewer #1: Yes

Reviewer #2: Yes

5. Is the manuscript presented in an intelligible fashion and written in standard English?

Reviewer #1: Yes

Reviewer #2: Yes

6. Review Comments to the Author

Reviewer #1: The authors have done an exhaustive work to address the comments raised by both referees and the revised version of the manuscript is much more solid and scientifically rigorous. Consequently, I think that the manuscript is now suitable for publication.

Nonetheless, I have a few minor comments regarding the modifications introduced by the authors:

- In my first revision, I thought that the authors were computing the effective reproduction number at time $t$ as the number of contagions made by a newly infected individual during his/her infectious period. In case that the authors define the effective reproductive number at time $t$ as the number of contagions made by an existing infectious individual at time $t$, which corresponds to the instantaneous reproductive number, its computation involves the past rather than the future of the dynamics. Therefore, the claim made in Page 4 on the relevance of the future dynamics for this quantity should be modified; nonetheless, both definitions are equivalent as long as the contact and recovery rates remain unchanged (see Nishiura, H., & Chowell, G. (2009). The effective reproduction number as a prelude to statistical estimation of time-dependent epidemic trends. In Mathematical and statistical estimation approaches in epidemiology (pp. 103-121).

- I like the new part of the discussion at the end of Page 15 on the modifications introduced by Knock et al in the publication following their preprint. Having said this, the description of the interaction between the authors of that paper and the authors of this manuscript are not matter of a scientific work and, therefore, should be removed, leaving possible further subjective interpretations to the reader.

- Regarding new Figure 6, I would use letters to label each panel to avoid describing them as a function of their position in the figure in the main text.

Reviewer #2: The authors addressed most of my previous comments. In particular, the inclusion of Figure 6 substantially improves the manuscript. Nevertheless, I still have some comments and technical details that are not clear to me.

The intention behind my comments regarding the impact of economic hardship was exactly on what you comment on in the new version of the manuscript. Temporal economic hardship is not the same as endemic/systemic poverty. Therefore, one should not expect the same impact on the health of individuals. I appreciate the inclusion of the two references on the indirect impact of COVID-19 and the restrictions that were put in place. However, I would like to highlight that neither of the references studies the impact of lockdowns in particular. Both article treat the indirect impacts of COVID-19 in general. The decrease in economic activity cannot be simply attributed to lockdown. The mere existence of a pandemic will affect economic activity, independently on whether restrictions are put in place or not. Furthermore, in the case of England, the closure of restaurants and leisure centres preceded lockdown. I recommend you to make the definition of lockdown more explicit. In particular, point out that the closure of restaurants and nightclubs is not included. As I understand, by lockdown you refer to the closure of non essential retail and the stay-at-home order that was announced on 23 March and took effect on 26 March.

In the reference on South Asia I did not find a part that would justify the conclusion that restrictions led to more deaths than the ones that have been prevented. The only estimation I can find are COVID-19 deaths in 2020 "if no additional mitigation strategies are instituted in the region this year”. However, in this case the mitigation measures, including for example the lockdown in India, are already included.

Nevertheless, I agree with the authors that the impact of lockdown on health besides COVID-19 should be taken seriously. I just ask the authors to be careful how they communicate this and how studies are interpreted.

I have a question regarding the incubation period. In section "3.1 - Corrections and minor modifications" the authors state that they use a mean duration of 5.8 days for the incubation period that is equivalent here with the time spent in compartment E. In contrast, in section "3.3 - Relaxing the model assumptions" the authors state that they shorten the E state to have an average of 3 days to infectivity. So which of both the authors actually consider? Or it this the difference between the plot in the top right and bottom left in Figure 6? Also, when the authors shorten the time in E to 3 days do they consider an additional compartment that represents pre-symptomatic infectiousness?

The authors comment on the published version of the manuscript by Knock et al. In particular, they mention that in the new version the reproduction number is below one in many regions before lockdown. Looking at their results, it seems that the reproduction number drops below one in many regions on the day the lockdown was announced. I think this is very consistent with the results you have here, even though the day of the announcement is not indicated in the graphics. Obviously, not only the implementation but also the announcement has an impact. This is definitely something you should comment on. I also recommend the authors to include a table where you explicit write when the reproduction number crosses one. It is somehow difficult to see this in the plot but is a crucial result of your analysis.

Additionally, I recommend to omit the comments regarding the exchange you had with the authors of Knock et al.. While it may be entertaining academic gossip, I do not think that it adds anything from a scientific perspective. At the end, the community needs to evaluate the content of manuscripts and not their creation process.

The figure 7 shows the reconstructed incidence. However, I would like to point out that if you assume a non exponential infection model, as it is the case for SARS-CoV-2 (approx. gamma distributed generation time), the reproduction number is not necessarily below one when the peak in infections is reached. As a matter of fact, depending on the decrease, the reproduction number will only drop below one some days afterwards. In this sense, if the peak is reached only a few days before lockdown, the figure you present is not very conclusive. You should comment on the limitation of using exponentially distributed waiting times.

My last comment is regarding the potential change in the evolutionary landscape of the virus that the lockdown may induce. As I understand, your argument applies to almost any measures that try to prevent the spread of the disease. I am thinking about social distancing or contact tracing for example. Accordingly, the conclusion would be to do nothing due to the fear of mutations? Or are there any other possible interventions that may contain the spread where this argument does not apply? I am not experienced in the biology of viruses so I cannot really judge the validity of your argument. However, I do not think it is necessary to motivate your study. I recommend omitting this comment as well as the appendix.

7. PLOS authors have the option to publish the peer review history of their article (what does this mean?). If published, this will include your full peer review and any attached files.

Reviewer #1: No

Reviewer #2: **Yes: **Benjamin Steinegger

---

## [Author Response · Author response to Decision Letter 1]

10 Aug 2021

Please see the attached response to reviews file.

---

## [Decision Letter · Decision Letter 2]

2 Sep 2021

Was R < 1 before the English lockdowns? On modelling mechanistic detail, causality and inference about Covid-19

PONE-D-21-08472R2

Dear Dr. Wood,

We’re pleased to inform you that your manuscript has been judged scientifically suitable for publication and will be formally accepted for publication once it meets all outstanding technical requirements.

Kind regards,

Alessandro Rizzo

Academic Editor

PLOS ONE

Additional Editor Comments (optional):

The paper is deemed as being in good shape for publication. I recommend the authors to apply the last suggestions by Reviewer 2 when submitting their final version. 

Reviewers' comments:

Reviewer's Responses to Questions

**Comments to the Author**

1. If the authors have adequately addressed your comments raised in a previous round of review and you feel that this manuscript is now acceptable for publication, you may indicate that here to bypass the “Comments to the Author” section, enter your conflict of interest statement in the “Confidential to Editor” section, and submit your "Accept" recommendation.

Reviewer #1: All comments have been addressed

Reviewer #2: (No Response)

2. Is the manuscript technically sound, and do the data support the conclusions?

Reviewer #1: Yes

Reviewer #2: Yes

3. Has the statistical analysis been performed appropriately and rigorously? 

Reviewer #1: Yes

Reviewer #2: Yes

4. Have the authors made all data underlying the findings in their manuscript fully available?

Reviewer #1: Yes

Reviewer #2: Yes

5. Is the manuscript presented in an intelligible fashion and written in standard English?

Reviewer #1: Yes

Reviewer #2: Yes

6. Review Comments to the Author

Reviewer #1: (No Response)

Reviewer #2: The authors addressed the points of my previous review. In this sense, I have only some minor comments:

- The reference from India (PIB India, 2020) is a press conference from the local authorities that was uploaded to YouTube. During the press conference, a power point presentation is shown with the estimations you reference here. From what I see, it seems not possible to look into the methodology that leads to their estimations. In this sense, I would be careful with the inclusion of these numbers. However, as you point in the text, I guess this corresponds to the estimations by the authorities in India.

- I appreciate the inclusion of the numbers as the reproduction number first was below one. However, I recommend the authors to include one example of the exact conversion between numbers and dates in the caption of Figure 6. Otherwise, the reader needs to scroll to Figure 1 for finding the conversion.

- The comments on the possible impact of lockdowns regarding the evolution of the virus seem reasonable to me. However, I would like to stress that I have no knowledge in virus evolution and can thus not judge plausibility.

-Please remove the undertone in the acknowledgements. It is not relevant on which basis the paper was rejected in PNAS. And, it is also not relevant whether you send Knock et al. an e-mail or not.

7. PLOS authors have the option to publish the peer review history of their article (what does this mean?). If published, this will include your full peer review and any attached files.

Reviewer #1: No

Reviewer #2: **Yes: **Benjamin Steinegger

---

## [Editor Report · Acceptance letter]

10 Sep 2021

PONE-D-21-08472R2 

Was *R* < 1 before the English lockdowns? On modelling mechanistic detail, causality and inference about Covid-19 

Dear Dr. Wood:

I'm pleased to inform you that your manuscript has been deemed suitable for publication in PLOS ONE. Congratulations! Your manuscript is now with our production department. 

Kind regards, 

on behalf of

Prof. Alessandro Rizzo 

Academic Editor

PLOS ONE